ecology/evolution

defence, *Chlorella vulgaris*, *Daphnia*, *Simocephalus*, predator–prey interaction

**Author for correspondence:**
Dania Albini
e-mail: dania.albini@swansea.ac.uk

# Turning defence into offence? Intrusion of cladoceran brood chambers by a green alga leads to reproductive failure

Dania Albini, Mike S. Fowler, Carole Llewellyn and Kam W. Tang

Department of Biosciences, Swansea University, Swansea SA2 8PP, UK

DA, 0000-0003-4236-1536; MSF, 0000-0003-1544-0407

Microalgae are the foundation of aquatic food webs. Their ability to defend against grazers is paramount to their survival, and modulates their ecological functions. Here, we report a novel anti-grazer strategy in the common green alga *Chlorella vulgaris* against two grazers, *Daphnia magna* and *Simocephalus* sp. The algal cells entered the brood chamber of both grazers, presumably using the brood current generated by the grazer's abdominal appendages. Once inside, the alga densely colonized the eggs, significantly reducing reproductive success. The effect was apparent under continuous light or higher light intensity. The algal cells remained viable following removal from the brood chamber, continuing to grow when inoculated in fresh medium. No brood chamber colonization was found when the grazers were fed the reference diet *Raphidocelis subcapitata* under the same experimental conditions, despite the fact that both algal species were readily ingested by the grazers and were small enough to enter their brood chambers. These observations suggest that *C. vulgaris* can directly inflict harm on the grazers' reproductive structure. There is no known prior example of brood chamber colonization by a microalgal prey; our results point to a new type of grazer–algae interaction in the plankton that fundamentally differs from other antagonistic ecological interactions.

## 1. Introduction

Aquatic primary production is dominated by planktonic microalgae [1]. Because of microalgae's small size, their limited to non-existent motility, and the lack of refuge in the open water, they are constantly exposed to grazing risk. Grazing pressure can be variable in space and time, and consequently, the ability of microalgae to efficiently defend themselves against grazing not

only affects their own survival, but also modulates the strength of trophic cascades and changes the outcome of competition in aquatic communities [2].

Broadly speaking, three types of defensive strategy are common among microalgae. A preventive strategy allows the cells to avoid ingestion, usually by forming colonies and spines to increase the organism's overall size beyond the grazer's handling capacity, or releasing chemical repellents and toxins to fend off the grazer [1,3,4]. A resistant strategy increases the cells' chance of post-ingestion survival, such as by having thicker cell walls [2,3]. Both preventive and resistant strategies work without inflicting direct harm on the grazers. A retaliatory strategy involves the production of intracellular toxic chemicals that would harm the grazers after ingestion [5,6]; this strategy, however, requires the sacrifice of the ingested cells to benefit the clonal populations. While there are examples in the animal kingdom where a prey takes the offensive against a would-be predator [7,8], there is no comparable example among the microalgae where a normal algal prey physically and actively attacks the grazer. Only one study [9] reported how the haptophyte *Prymnesium parvum* was able to release toxins that lysed the predator *Oxhyrris marina* cells, and to subsequently phagocyte the remains of the predator cells. However, the prey strategy can be considered as a chemical defence, rather than a physical attack. The lack of information can be partially attributed to the fact that it is counterintuitive to consider that a normal microalgal prey, which may be an order of magnitude smaller in size than its grazer, has the ability to attack the grazer. It is difficult to measure and recognize prey attacks and to distinguish them from a retaliatory response. Here, we report a novel phenomenon where a microalga turns defence into offence by attacking the grazer's reproductive structure without ingestion, leading to reproductive failure in the grazer.

Cladocerans are among the most prevalent planktonic grazers in freshwater systems, feeding mainly on microalgae of 1–50 µm in size [10,11]. The movement of their thoracic appendages creates a water current that carries the microalgae into the filter chamber that is formed by the valves of the carapace [12,13]. Microalgae are then transported to the food groove [11]. Cladocerans' high feeding rates and prolific reproduction can exert strong top-down control of the algal populations, creating a clear-water phase in many lakes [14–16]. Under optimal conditions, they reproduce via parthenogenesis where mature females carry eggs in their brood chambers; the eggs develop until they emerge as fully formed individuals. The production of eggs that fail to develop into viable offspring is a wasted investment of energy and resources and it leads to a decline in fitness. Given that conditions within the brood chamber and the surrounding medium are similar [17], we might expect to find algae entering the brood chamber via the brood current.

We discovered a novel anti-grazer offensive trait in the freshwater alga *Chlorella vulgaris* (Chlorellaceae) against the cladoceran grazers *Daphnia magna* and *Simocephalus* sp. *Chlorella* spp. are commonly found in ponds and lakes, and are readily ingested by cladoceran grazers [10,18,19]. There are no prior reports of *C. vulgaris* exerting harmful effects on cladoceran grazers; quite to the contrary, it is frequently used as a reference diet in ecotoxiciological studies [18,19]. We initially conducted two experiments to determine what light conditions and algal densities were associated with *Daphnia* brood chamber colonization. *Chlorella vulgaris* has a global distribution, stretching from the arctic [20] to the tropics [21], which suggests that the photoperiods and light intensities used in our experiments will be found in their natural range. We went on to investigate what proportion of grazers were colonized in this way, how many grazer offspring developed and whether the algae that colonized the brood chambers remained viable following the intrusion. We conducted a third experiment to determine the generality of our basic findings in another cladoceran grazer, *Simocephalus* sp. We observed that *C. vulgaris* cells were able to enter the brood chamber, colonize the eggs and cause reproductive failure in both grazer species. To our knowledge, this is the first report of harmful intrusion of cladoceran brood chambers by a freshwater microalga, which represents a new type of predator–prey interaction in the plankton.[1]

# 2. Material and methods

## 2.1. Organisms

An inoculum of *C. vulgaris* (Chlorellaceae) was obtained from the Culture Collection of Algae and Protozoa (CCAP, National British service culture collection), strain 211/11B. *Chlorella vulgaris* has

---

[1]We conducted a literature search using the following databases: Google, Ecosia, Research Gate, Google Scholar, Safari, Scopus, Direct Science; and various combinations of keywords: Cladocera, colonized, colonised, algae, internal tissue, zooplankton, brood chamber, eggs, Chlorella, Daphnia. No report of this or similar phenomenon could be found.

spherical-shaped cells of 2–10 µm in diameter. It was cultured in BG-11 medium (Sigma-Aldrich, 73816 FLUKA) at $20 \pm 1°C$ under an 18 L : 6D photoperiod and a light intensity of 70 µmol photons $m^{-2} s^{-1}$. *Raphidocelis subcapitata* (Selenestraceae) (8–12 µm in length and 2–3 µm in width) is commonly used as food for *D. magna* in ecotoxicological studies, and was included in this study as a reference diet. *Raphidocelis subcapitata* inoculum was obtained from CCAP (strain no. 278/4), and was cultured under the same conditions as *C. vulgaris*.

The *D. magna* (Cladocera) used in this study was provided by the Leibniz-Institute of Freshwater Ecology and Inland Fisheries. Genetically identical females were grown at $20 \pm 1°C$, under an 18 L : 6 D photoperiod and a light intensity of 70 µmol photons $m^{-2} s^{-1}$. *Daphnia magna* was fed ad libitum daily with an equal mixture of *C. vulgaris* and *R. subcapitata*. *Simocephalus* sp. (Cladocera) was originally collected from a pond in Upper Killay in Wales, UK (51°36′46.5″ N, 4°02′38.5″ W). Genetically identical females were grown in the same conditions as *D. magna*.

## 2.2. Experiment 1 (electronic supplementary material, figure S1)

Genetically identical *D. magna* newborns (i.e. less than 24 h old) were isolated from the stock, fed a mixture of *C. vulgaris* and *R. subcapitata*, and used for experiments when they were 5 days old and of mean body size $3.00 \pm 0.45$ mm s.e., with empty brood chamber. The experiment consisted of different combinations of low $(1 \times 10^6 \text{ cells ml}^{-1})$ and high $(6 \times 10^6 \text{ cells ml}^{-1})$ concentrations of *C. vulgaris* plus a reference diet of *R. subcapitata* $(1 \times 10^6 \text{ cells ml}^{-1})$, under two photoperiods (24 L and 18 L : 6D; light intensity 70 µmol photons $m^{-2} s^{-1}$). These experimental food concentrations were well within the natural algal concentrations encountered by cladocerans [22]. The experiments were conducted in 20 ml test tubes containing Evian spring water (pH = 7.2; $Ca^{2+} = 78$ mg $l^{-1}$) and nutrients (BG-11 in 1 : 100 dilution), plus one of the algal species-concentration combinations for a final volume of 17 ml in each replicate. The tubes were tapped with a breathable film in order to allow the exchange of oxygen but to limit the evaporation of the water.

Before the experiment, each *D. magna* female was cleaned with water to remove any trace of food. One *D. magna* was added to each test tube. Twenty replicates of each treatment were run at $20 \pm 1°C$, for 20 days. The test tubes were gently inverted twice daily to avoid algal sedimentation and rotated in position to expose them to equal illumination throughout the experiment. Each *D. magna* was checked daily for the presence of eggs in the brood chamber and egg development time. Newborns were counted and removed from the test tubes. A magnifying lens (40× 25 mm magnification) was used first to check for the presence of algae inside the brood chamber; on Day 0, 3, 6, 9 and 12, the females were transferred onto a glass slide and examined more closely for brood chamber colonization under a microscope, and photographs were taken. This process usually took less than 10 min and the animals were returned afterward to the test tubes unharmed. Dead *D. magna* were checked and removed from the experiment; *C. vulgaris* could still be seen inside the brood chamber and appeared green 2 days after the death of the grazer. The algal suspensions were renewed every 5 days to prevent food shortage for the grazers. This procedure took *ca* 2 h and was always performed at the same time of day.

## 2.3. Experiment 2 (electronic supplementary material, figure S2)

The second experiment was conducted to test for brood chamber colonization under different light intensities (70 and 130 µmol photons $m^{-2} s^{-1}$; photoperiod 18 L : 6 D). *Daphnia magna* was raised under the same conditions described for Experiment 1. *Chlorella vulgaris* and *R. subcapitata* were added at a concentration of $6 \times 10^6 \text{ cells ml}^{-1}$. *Daphnia magna* was checked daily for the same parameters as in Experiment 1. The experiment lasted 20 days; the algal suspensions were renewed every 5 days to prevent food shortage for the grazers. *Daphnia magna* showing signs of brood chamber colonization were examined more closely under a microscope and photographed. At the end of the experiment (Day 20), females with *C. vulgaris* in their brood chamber were killed and washed externally before the brood chamber was cut open to release the algal cells from within, with algae then transferred using a sterile syringe into individual test tubes. A total of 16 test tubes with algae retrieved from 16 *D. magna* individuals were filled with culture medium (up to 7 ml) composed of autoclaved deionized water and BG-11 nutrients. On Day 0 (day of inoculation of the algae retrieved from the brood chambers, in culture medium), Day 3 and Day 5, a 1 ml aliquot was transferred from each test tube to an Eppendorf tube with a drop of Lugol's solution (Sigma-Aldrich 62650-1 L-F) added as fixative. The fixed samples were kept in a dark refrigerator until counted, within 5 days, with a haemocytometer under a Leica inverted microscope (400× magnification).

## 2.4. Experiment 3

The third experiment was conducted to test for brood chamber colonization in another cladoceran species, *Simocephalus* sp. Genetically identical females of less than 24 h old were isolated from the stock, fed a mixture of *C. vulgaris* and *R. subcapitata*, and used for experiments when they were 5 days old with empty brood chambers. Both algae were cultured in BG-11 medium (Sigma-Aldrich, 73816 FLUKA) at $20 \pm 1°C$ under an 18 L : 6 D photoperiod and a light intensity of 70 µmol photons $m^{-2} s^{-1}$. This experiment consisted of two beakers each with 40 *Simocephalus* individuals, and *C. vulgaris* or *R. subcapitata* was added at a concentration of $1 \times 10^6$ cells $ml^{-1}$. Evian spring water (pH = 7.2; $Ca^{2+}$ = 78 mg $l^{-1}$) and nutrients (BG-11 in 1 : 100 dilution) were added to a final volume of 1 l. The beakers were exposed to a photoperiod of 18 L : 6 D of 130 µmol photons $m^{-2} s^{-1}$ at a temperature of $20 \pm 1°C$, and were gently mixed twice daily to avoid algal sedimentation. A magnifying lens was used to check daily for the presence of algae inside the grazer's brood chamber. On Days 0, 3, 6 and 7 a subsample of grazers was transferred onto a glass slide and examined more closely under a microscope and photographed in the same manner as in Experiment 1. Newborns were checked and removed daily. Females with eggs and algae in their brood chamber were isolated into a new beaker and fed the same algal species, to follow the fate of the eggs. The experiment lasted 10 days.

## 2.5. Data analysis

Statistical analysis was performed with R-studio v. 1.1.419. An additive, factorial generalized linear model (GLM) with a binomial error family tested for the effect of experimental treatments (algal species, initial concentration, photoperiod and/or light intensity) on the probability of colonization by the algae. The effect of *C. vulgaris* colonization on the reproductive output of the grazer was assessed using an additive factorial negative binomial GLM [23] examining the effects of the same factors listed above on the number of newborns at the end of the experiment. Given the lack of interaction terms in our analyses (in some cases due to lacking a fully crossed experimental design), we did not perform further pairwise comparisons on these statistical models. Each factor in the additive GLMs consists of only two levels (two photoperiods, two light intensities, two algal species, two initial algal concentrations), therefore, any significant differences associated with a factor can be interpreted as a difference between those two factor levels. Inclusion of interaction terms, where appropriate, did not result in better-performing models in any case (based on AICc comparison), or qualitatively change our interpretations.

# 3. Results

## 3.1. Experiment 1

The grazer *D. magna* ingested both algal species, *C. vulgaris* and the reference diet *R. subcapitata*, as confirmed by its full gut (figure 1). However, some *C. vulgaris* cells were found inside the *D. magna* brood chamber (figures 1 and 2) where they disrupted the egg development (figure 2 and table 1). While a few *D. magna* females died before colonization occurred, *C. vulgaris* cells were clearly visible inside the brood chambers of most of the surviving individuals kept under constant light (figure 2). In some instances, the eggs were heavily covered by the algae (figure 1*b*), and algae remained inside for the remainder of the experiment. Brood chamber colonization did not occur under the 18 L : 6 D photoperiod with *C. vulgaris*, or with *R. subcapitata* in either photoperiod, i.e. there were significant effects of photoperiod and algal species, but not initial algal concentration, on the probability of *Daphnia* brood chamber colonization (table 1 and figure 2). Colonization was first observed under constant light conditions between Days 4 and 5 (mean $4.7 \pm 0.40$ s.e.) with high algal concentration and between Days 5 and 6 (mean $5.6 \pm 0.35$ s.e.) with low algal concentration.

   Similar patterns arose for *Daphnia* reproduction, with significant effects of photoperiod and algal species, but no effect of algal concentration, on the number of newborn *Daphnia* (table 2 and figure 2). Overall, 50–67% of *D. magna* in *R. subcapitata* treatments, and 31–65% of *D. magna* in *C. vulgaris* treatments under the 18 L : 6D photoperiod produced viable eggs (i.e. eggs that developed into newborns). In comparison, only 0–20% of *D. magna* produced viable eggs in *C. vulgaris* treatments under continuous light (figure 2).

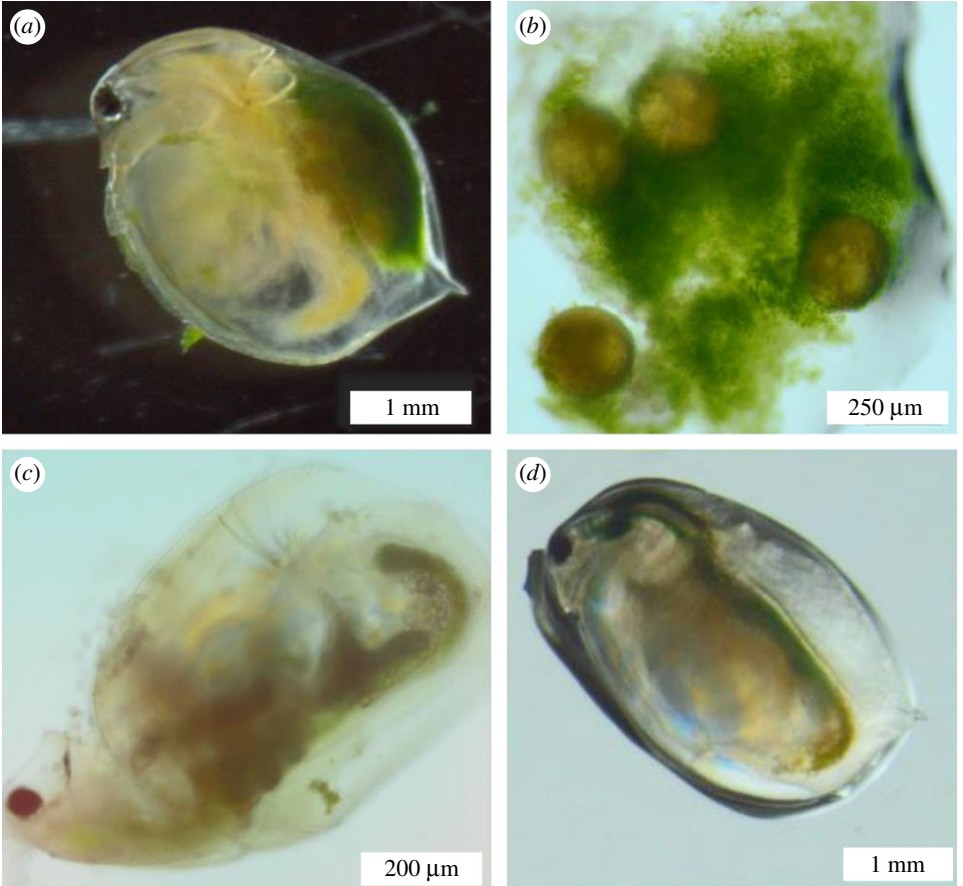

**Figure 1.** Colonization of brood chamber by *C. vulgaris* as observed under the microscope. (*a*) *Daphnia magna* with eggs colonized by *C. vulgaris* in the brood chamber (Experiment 1). (*b*) Eggs and *C. vulgaris* cells extracted from the brood chamber (Experiment 1). (*c*) A female *Simocephalus sp.* with *C. vulgaris* in the brood chamber (Experiment 3). (*d*) *Daphnia magna* showing no body colonization with the reference diet *R. subcapitata* (Experiment 2).

## 3.2. Experiment 2

We tested for brood chamber colonization by exposing *D. magna* to *C. vulgaris* and *R. subcapitata*, both at a concentration of $6 \times 10^6$ cells ml$^{-1}$, under two different light intensities of a single photoperiod (18 L : 6D). Under the higher light intensity (approx. 130 µmol photons m$^{-2}$ s$^{-1}$), *C. vulgaris* colonized the brood chamber of *D. magna* and resulted in significantly fewer newborns (table 3 and figure 3). Colonization first occurred on Day 3, when *C. vulgaris* cells were clearly visible inside the brood chamber. Many of the eggs were densely covered by the algae. The mean time for colonization to occur was 3.5 days (±0.18 s.e.) from the start of the experiment. Brood chamber colonization did not occur under the lower light intensity (approx. 70 µmol photons m$^{-2}$ s$^{-1}$; 18 L : 6 D), or in either of the *R. subcapitata* treatments (figure 3).

There was a significant effect of light intensity on the number of *D. magna* newborn, with fewer offspring born under the higher light intensity (table 4 and figure 3). Overall, under the lower light intensity, 95% of *D. magna* in *R. subcapitata* treatment and 90% of *D. magna* in *C. vulgaris* treatment produced viable eggs. Under the higher light intensity, only 5% of *D. magna* treated with *C. vulgaris* produced viable eggs, whereas all females treated with *R. subcapitata* (100%) produced newborn.

*Chlorella vulgaris* cells extracted from the brood chamber of *D. magna* appeared green and alive under the microscope. After inoculation in fresh medium, the cell abundance increased from *ca* $2 \times 10^5$ cells ml$^{-1}$ to *ca* $4 \times 10^5$ cells ml$^{-1}$ over 5 days, confirming that the cells remained viable.

## 3.3. Experiment 3

To confirm the generality of our main finding, we tested for brood chamber colonization with another cladoceran, *Simocephalus* sp. When fed with the *R. subcapitata* diet, *Simocephalus* sp. started to produce

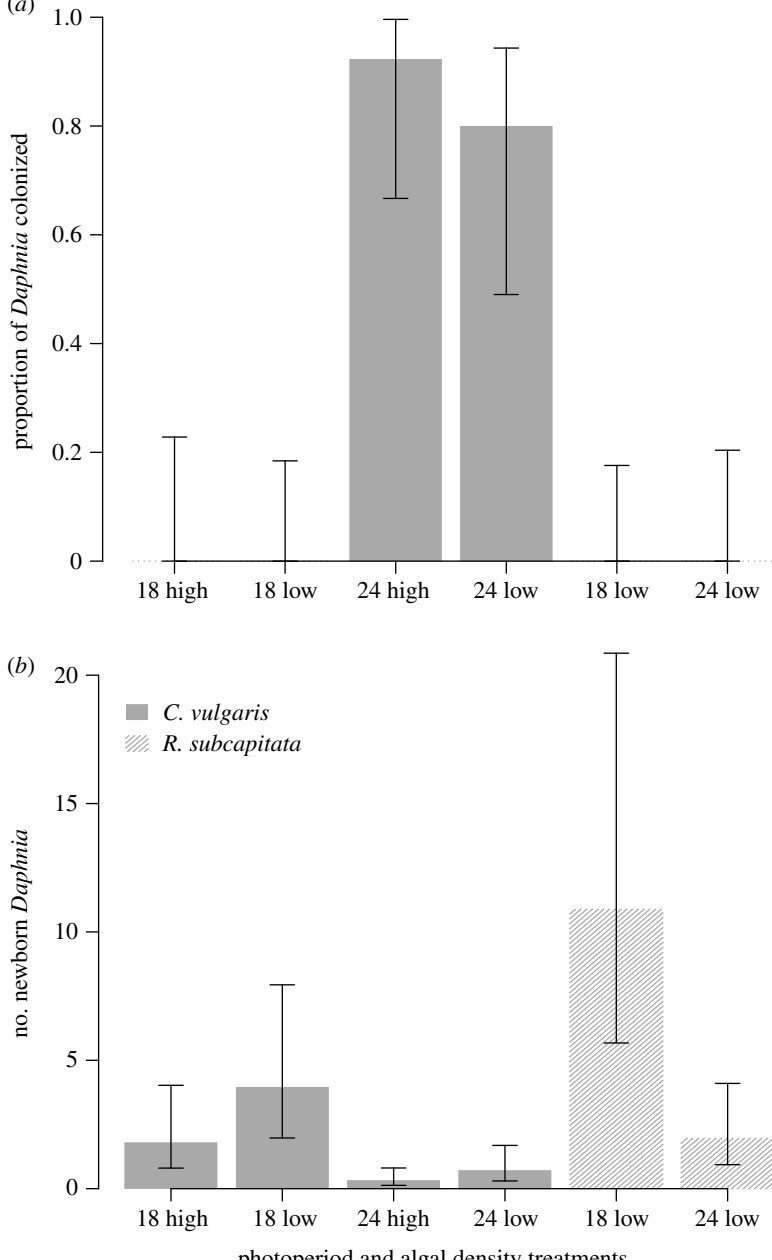

**Figure 2.** Colonization of *D. magna* brood chambers under an extended photoperiod leads to a reduction in reproductive output. Panel (*a*) shows the proportion (±95% binomial CIs) of *Daphnia* that were colonized and (*b*) shows the daily number of newborn *Daphnia* (mean ± 95% CIs from a negative binomial GLM), when fed with one of two algal species (*C. vulgaris*, and *R. subcapitata*) under low (1 × 10⁶ cells ml⁻¹) or high initial algal concentrations (6 × 10⁶ cells ml⁻¹) and light conditions (18 : 6 or 24 : 0 h light : dark) in Experiment 1 (17 ≥ *n* ≤ 20 *Daphnia*, kept individually, survived in each treatment). Data are shown for the overall experimental time.

newborns on Day 7 with a mean number of 4.5 ± 0.3 s.e. newborns per individual per day. By contrast, the *C. vulgaris* treatment resulted in zero newborn by the end of the experiment (Day 10).

Chlorella vulgaris colonization of the *Simocephalus* brood chamber occurred on Day 7 in 33 out of 40 females (figure 1*c*). Six females had their eggs covered by the algae (probability = 0.18, 95% binomial CI = [0.07, 0.35]), while the remaining 27 females had not yet deposited eggs, but did have algae in the brood chamber. These six females were isolated for further observations, and all died after 4 days with no newborns produced. Brood chamber colonization did not occur in the *R. subcapitata* treatment.

**Table 1.** Analysis of deviance based on the binomial GLM of the proportion of *D. magna* brood chambers colonized at the end of Experiment 1 by the two green algae *R. subcapitata* and *C. vulgaris* (algal species) under two different photoperiods (18 : 6 versus 24 : 0 light : dark cycle) and initial algal densities (concentration: $1 \times 10^6$ versus $6 \times 10^6$) treatments ($n \leq 20$ for each treatment).

| experimental factor | d.f. | deviance | resid. d.f. | resid. dev. | Pr(>Chi) |
|---|---|---|---|---|---|
| null | | | 86 | 199.22 | |
| photoperiod | 1 | 66.65 | 84 | 52.58 | <0.001 |
| algal species | 1 | 34.76 | 83 | 17.81 | <0.001 |
| concentration | 1 | 0.75 | 82 | 17.06 | 0.386 |

**Table 2.** Analysis of deviance based on negative binomial GLM of the total number of newborns produced by *D. magna* under different experimental treatments in Experiment 1. Factor levels are as described in table 1.

| experimental factor | d.f. | deviance | resid. d.f. | resid. dev. | Pr(>Chi) |
|---|---|---|---|---|---|
| null | | | 84 | 104.35 | |
| photoperiod | 1 | 13.02 | 83 | 91.33 | <0.001 |
| algal species | 1 | 11.92 | 82 | 79.41 | <0.001 |
| concentration | 1 | 2.47 | 81 | 76.94 | 0.116 |

**Table 3.** Analysis of deviance based on a binomial GLM of the proportion of *D. magna* brood chambers colonized by the two green algae *R. subcapitata* and *C. vulgaris* (algal species) under two different light intensities (130 versus 70 µmol photons $m^{-2} s^{-1}$) in Experiment 2 (18 : 6 photoperiod, $n \leq 20$ for each treatment).

| experimental factor | d.f. | deviance | resid. d.f. | resid. dev. | Pr(>Chi) |
|---|---|---|---|---|---|
| null | | | 75 | 78.23 | |
| light intensity | 1 | 27.61 | 74 | 50.62 | <0.001 |
| algal species | 1 | 43.01 | 73 | 7.61 | <0.001 |

## 4. Discussion

We have observed a novel, surprising phenomenon where the green microalga *C. vulgaris* repeatedly colonized the brood chamber of two cladoceran grazer species under increased light availability, leading to reproductive failure. The grazers were fed a mixture of *C. vulgaris* and *R. subcapitata* for 5 days until the females were reproductively ready (formation of brood chamber) before the experiments; therefore, we may rule out the likelihood that any subsequent reproductive failure was caused by the grazers' pre-conditions. Indeed, reproductive success remained high in the *R. subcapitata* treatments and the 18 : 6 photoperiod and lower light intensity *C. vulgaris* treatments, across all three experiments.

The dense cover of algal cells may have restricted the transfer of oxygen and nutrients to the cladoceran eggs, leading to the eggs' premature death. Our observations raise the question: how did *C. vulgaris* cells enter the brood chamber? The cladoceran's body is enclosed by a carapace; the brood chamber is separated from the external environment by the first abdominal process (a structure in the lower rear part of the abdomen), which normally 'closes' the bottom part of the brood chamber [19]. Upward movements of the abdomen and thoracic appendages open the abdominal processes and create a pumping action to pump water through the posterior into the brood chamber and out through small gaps in the ventral carapace. This 'brood current' is important for supplying oxygen to the eggs [13], and in our case is probably responsible for bringing *C. vulgaris* cells into the brood chamber. Both *C. vulgaris* and *R. subcapitata* cells were less than 12 µm in the longest dimension, whereas the gap when *D. magna* and *Simocephalus* sp. opened their abdominal processes was 577 ± 7.05 µm (mean ± s.d., $n = 50$; electronic

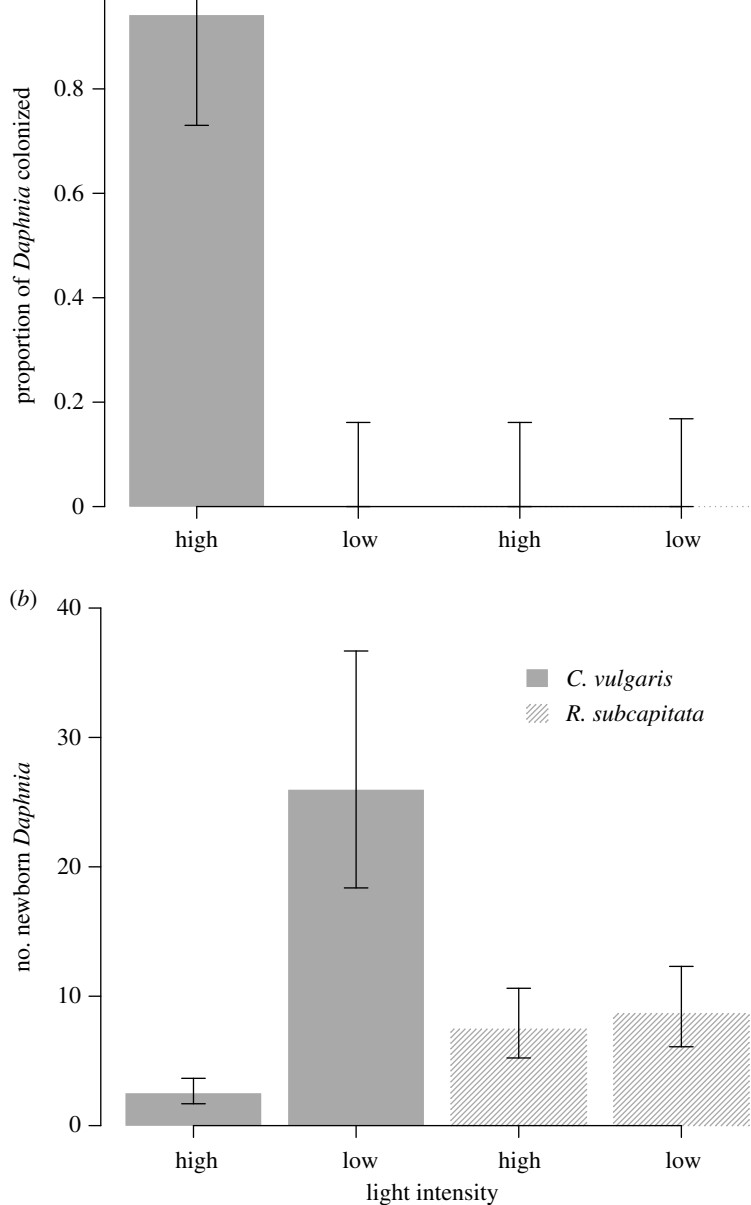

**Figure 3.** Colonization of *Daphnia magna* brood chambers under higher light intensity leads to a reduction in reproductive output. Panel (*a*) shows the proportion (±95% binomial CIs) of *Daphnia* that were colonized and (*b*) shows the daily number of newborn *Daphnia* (mean ± 95% CIs from a negative binomial GLM), when fed with one of two algal species (*C. vulgaris*, and *R. subcapitata*) under low (70) or high (130 μmol photons m$^{-2}$ s$^{-1}$) light intensities in Experiment 2. (17 ≥ *n* ≤ 20 *Daphnia*, kept individually, survived in each treatment). Data are shown for the overall experimental time.

**Table 4.** Analysis of deviance based on a negative binomial GLM of the total number of newborns produced by *D. magna* under different experimental treatments in Experiment 2. Factor levels are described in table 3.

| experimental factor | d.f. | deviance | resid. d.f. | resid. dev. | Pr(>Chi) |
|---|---|---|---|---|---|
| null | | | 78 | 134.32 | |
| light intensity | 1 | 12.28 | 77 | 122.04 | <0.001 |
| algal species | 1 | 20.58 | 76 | 101.46 | <0.001 |

supplementary material, figure S3) and $500 \pm 33\ \mu m$ (mean $\pm$ s.d., $n = 50$), respectively. Hence, the gap was clearly large enough for both algal species to enter the brood chamber via the brood current. Indeed, there has even been a report of predatory copepods entering *Daphnia* brood chamber to prey on the eggs [13]; it is no surprise that microalgae could enter the brood chamber as well. Nevertheless, in our experiments only *C. vulgaris* colonized the brood chamber, while *R. subcapitata* did not, under the light conditions and food concentrations we considered. While we did not fully cross all possible algal concentration combinations in Experiment 1 (due to practical and resource limitations), we did explicitly test the higher *R. subcapitata* concentrations $(6 \times 10^6\ \mathrm{cells\ ml}^{-1})$ in Experiment 2 and still did not record any brood chamber colonization with this algal species. For this reason, we believe our general interpretations to be robust. This suggests that our observations were not a result of the grazers 'drowning in food', but rather that *C. vulgaris* had specific properties that enabled colonization. Venancio *et al*. [24] showed that *C. vulgaris* and *R. subcapitata* are very different in their abilities to survive under different environmental factors. In fact, the former can survive under a wider range of ionic conditions compared with the latter [24]. Therefore, *C. vulgaris* has previously been shown to have a competitive advantage over *R. subcapitata*.

The ability of the *C. vulgaris* cells to remain inside the brood chambers and attach to the eggs suggests that the cells had adhesive property. Many microalgal species can produce exopolymeric substances (EPS) that act as bio-adhesives [25]. Among the green microalgae, *Chlorella* species are particularly known for their high production of EPS under specific growth conditions, which is exploited for biotechnological applications [25]. Several experimental studies have reported higher EPS production by microalgae under higher light availability [25–28]. This is consistent with the idea that EPS production is a result of excess photosynthetic carbon fixation [27]. Following this reasoning, we may postulate that light is a limiting resource that affects EPS production, which in turn determines the success of brood chamber colonization. Indeed, in our study, brood chamber colonization by *C. vulgaris*, both in terms of the time for colonization to occur and the subsequent failed egg development, was more severe under higher light exposure (either light intensity or photoperiod).

Although *C. vulgaris* was readily eaten with no discernible effect on the survival of the grazers over the timespan of our experiments, algal colonization of the brood chamber had a very clear and detrimental effect on reproductive success. This observation sets this trait apart from the other common anti-grazer traits in microalgae, such as colony formation, spine formation and chemical repellents, which work by relieving grazing pressure from the prey cells, but the grazers (or their offspring) are not necessarily harmed in the process. By contrast, *C. vulgaris* colonization of the brood chamber is akin to turning defence into offence by directly attacking the grazer's reproductive structure and disrupting egg development, often causing total reproductive failure. This phenomenon should not be confused with parasitism. The most commonly encountered parasites in cladocerans are bacteria and microsporidia [11], many of which can negatively impact the zooplankton's fecundity and survival [29]. Field studies have reported negative effects of parasites on the presence of eggs in the brood pouch of infected, in comparison with uninfected females [11,29]. However, our observations are fundamentally different from host–parasite interactions because parasites—by definition—are organisms that have evolved to inflict harm on the hosts, and they are not purposefully pursued by the hosts as 'food'. By contrast, *C. vulgaris* is actively grazed and this species is widely used as food in zooplankton cultivation and feeding experiments [10,30]. Our discovery, therefore, points to a previously unrecognized algal defensive trait and it suggests rethinking of the classical view of prey–predator interactions among the plankton.

Another important observation is that *C. vulgaris* cells that had colonized the brood chamber remained viable and were subsequently able to grow in fresh culture. Hence, these *C. vulgaris* cells were potentially able to harm the grazers without exposing themselves to certain mortality. The fact that this trait has not been shown to be widespread among microalgae suggests there is likely be a substantial (as yet unknown) cost associated with it. Even when the algal cells remain viable inside the brood chamber, unless the cells escape the brood chamber, they will still be removed from the population when the grazer is eaten by higher predators or settles out of the photic zone after death. Further research into the fate of the algal cells will help us understand the algal population dynamics and the cost associated with brood chamber colonization. Microsensors or chemical dyes may be used to monitor the interior environment of the brood chamber and explain the cause of reproductive failure. A mature cladoceran female produces multiple clutches during her lifetime. While our experiments show a clear negative effect of *C. vulgaris* on the existing clutch, it remains to be investigated whether the grazer can recover from brood chamber colonization and resume reproduction, which is critical for understanding the long-term effect on the grazer populations.

This new defensive mechanism against high exposure to light intensity and predation can be very important in the natural aquatic ecosystems. In fact, it is well known that climate changes are playing

a major role in the modification of aquatic bodies. Altered temperatures, thermocline depths, light penetrations and nutrient inputs would be expected [31]. Increased light exposure can affect the rates of phytoplankton primary production and species composition would change, annual production would increase and phytoplankton biomass might increase, producing cascade effects on the zooplankton abundance and on their interactions.

## 5. Conclusion

Cladocerans are the linchpin of the aquatic food web in lakes and reservoirs. The ability of microalgae to invade their brood chamber not only improves the algal species' own chances of survival, but also reduces the grazer's fitness, modulating the strength of trophic cascades within the food web. This novel discovery is very different from other anti-grazer traits among microalgae, and is also fundamentally different from parasitism. Our study, therefore, highlights a previously unknown type of grazer–algae interactions that can have significant ramifications at the population, community and ecosystem levels and in both natural and applied settings. No previous literature show comparable experiments conducted with *Chlorella* and *Daphnia*, and we believe our study will stimulate further similar investigations.

Data accessibility. The data are published in Dryad: https://dx.doi.org/10.5061/dryad.0p2ngf1x9 [32].

Authors' contributions. D.A. and K.W.T. conceived the idea of the study; D.A., M.S.F. and K.W.T. designed the experiments; D.A. conducted the experiments; D.A., M.S.F. and K.W.T. analysed the data; D.A., M.S.F. and K.W.T. drafted the manuscript with input from C.L. All authors gave final approval for publication.

Competing interests. We have no competing interests.

Funding. D.A. and C.L. received support from a Swansea University scholarship and the Phyconet project (Grant code: BB/L013789/1). K.W.T. and M.F. did not have funding support.

Acknowledgements. We thank Vanessa Ndovela for assistance during the experiments.

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
