## [Reviewer comments · Royal Society Open Science]

Review History

RSOS-200249.R0 (Original submission)

Review form: Reviewer 1

Is the manuscript scientifically sound in its present form?

Yes

Are the interpretations and conclusions justified by the results?

Yes

Is the language acceptable?

Yes

Do you have any ethical concerns with this paper?

No

Have you any concerns about statistical analyses in this paper?

No

Recommendation?

Accept with minor revision (please list in comments)

Comments to the Author(s)

This is an interesting work. Adaptations of algae and other phytoplankton groups against grazers such as cladocerans are long known. These include formation of colonies, development of spines, elevated levels of toxin production, thickened cell wall formation, etc. Even without apparent morphological adaptations, simple and non-colonial alga such *Chlorella* can still develop resistance against grazers (Yoshida et al. 2004 Proc. R. Soc. Lond. B 271: 1947–1953). The authors have shown *Chlorella* colonizes the brood chamber/pouch of cladocerans to kill the grazers. This observation as such is not novel. It is known that the brood chamber of cladocerans is of two types: open and close. For many species of cladocerans including *Daphnia*, the brood pouch is open posteroventrally, where it is contact with the medium. From this region, epibiotic microorganisms or those from the medium enter the brood chamber (Smirnov NN 2018. Physiology of the Cladocera 2nd edition, Academic Press). In this case the authors have observed that *Chlorella* enters the brood pouch. The growth of algae in the cladoceran brood pouch is also not entirely unexpected because the medium in the brood pouch maintains osmotic concentration similar to that of the surrounding medium (Aladin & Potts 1995. J Comp Physiol B 164: 671-683). Since the conditions in the brood pouch are not hostile to alga or bacteria, fungi, etc., such organisms can proliferate there.

Review form: Reviewer 2

Is the manuscript scientifically sound in its present form?

Yes

Are the interpretations and conclusions justified by the results?

Yes

Is the language acceptable?

Yes

Do you have any ethical concerns with this paper?

No

Have you any concerns about statistical analyses in this paper?

No

Recommendation?

Major revision is needed (please make suggestions in comments)

Comments to the Author(s)

The authors did an interesting job. They analyzed the anti-grazer strategy in the green alga *Chlorella vulgaris* against two cladoceran grazers, *Daphnia magna* and *Simocephalus* sp. They found that *C. vulgaris* could directly inflict harm on the reproductive structure of *D. magna*, and the colonization process depended on light conditions. To our knowledge, this is indeed the first study demonstrating the harmful intrusion of cladoceran brood chambers by a freshwater microalga, although the underlying mechanism and the specific process has still not been well clarified in the present study. Therefore, I recommend publishing this novel discovery in the journal after a major revision. The specific opinions are as follows.

Major questions

Method section

(1) It is unclear whether this phenomenon could occur in naturally aquatic ecosystems given that the colonization of brood chamber by *Chlorella vulgaris* mainly depended on extremely light

conditions. Firstly, the photoperiod throughout their experiments were 18L:6D or even 24 L: 0D, which was not in accordance with the natural condition. This is particularly true when the long-term light intensity ($130 \mu\text{mol photons m}^{-2} \text{ s}^{-1}$) was also considered.

(2) Although with similar cell size, the morphology of the two used algal species was different, is this the reason for their difference in colonization ability? This possibility should be discussed.

(3) The third experiment lasted for 10 days. Is it long enough for individuals of *Simocephalus* to reach their maximum body size? If not, your conclusion that *Simocephalus* individuals cannot be colonization by *Chlorella vulgaris* is not rigorous. Please explain.

Discussion section

(4) You suggest that the success of the *C. vulgaris* cells to remain inside the brood chambers was associated with its ability in high production of EPS under higher light availability? How do you know *R. subcapitata* do not have this ability? In other words, why *R. subcapitata* cannot invade grazers brood chamber?

(5) In addition, the underlying mechanisms for the failure of *C. vulgaris* cells to invade brood chamber of *Simocephalus* should also be discussed.

(6) Any microbial process facilitate the intrusion?

Minor questions.

1. Line 4-7, Page 3, the experimental procedure is not clear. For example, the 20 ml tube contain 17 ml Evian spring water, is oxygen availability of cladoceran grazers limited by this treatments? Was the tube capped with breathable polyethylene?

2. Line 13-14, page 4, What kind of magnifying lens was used? Is the magnification large enough to do this work?

3. Line 36-37, page 4, The culture condition for algae should be added here.

4. Line 1-5, Page 5, Did you check the prerequisites for GLM tests?

5. Line 20-38, page 6, It would be better if you could give some pictures to prove your hypothesis about the specific process how *C. vulgaris* cells entrance the brood chamber of *D. magna*.

6. Line 20-38, page 6, Is it possible for *C. vulgaris* cells entrance the brood chamber of *D. magna* from the postabdomen of *D. magna*, such as during the process of discharging neonates.

7. For figure 2 and 3, it is not clear what data you used for these two figures. Is it at the end of the experiment on a specific day?

Review form: Reviewer 3

Is the manuscript scientifically sound in its present form?

Yes

Are the interpretations and conclusions justified by the results?

Yes

Is the language acceptable?

Yes

Do you have any ethical concerns with this paper?

Yes

Have you any concerns about statistical analyses in this paper?

Yes

Recommendation?

Accept with minor revision (please list in comments)

Comments to the Author(s)

Review of Manuscript RSOS-200249

General comments

The results presented in this paper show the ability of one common species of microalga (*Chlorella vulgaris*) to colonise the brood chambers and induce reproductive failure in two cladoceran species. One of them, of great significance in the ecosystems and the scientific literature (*Daphnia magna*). Colonization of the brood chamber is positively related with longer light-photoperiod and stronger light intensity. There is some issue with the experimental design, that authors should solve, and its detailed in the specific comments below.

I miss some discussion about an ecological interpretation (beyond that high light conditions increase EPS production by *C. vulgaris*) of the potential significance of this mechanism of stronger exposure to light in the real ecosystems.

The authors claim that this is the first time that an interaction of this nature was found. However, something similar was found between two unicellular planktonic organisms: the haptophyte *Prymnesium parvum* and the dinoflagellate *Oxhyrris marina* (Tillmann et al. 2003, Aquatic Microbial Ecology 34: 73-84). This dinoflagellate is a heterotrophic organism that usually phagocytes phytoplankton cells, like *P. parvum*. *P. parvum* was observed to release cytolytic toxins that lyse *O. marina* cells, to phagocyte the small remains of the cells. This interaction is different that the one reported here because the predator actually becomes the prey, however, I think it needs to be referenced.

I would recommend this paper for publication after the authors solve the few questions presented here.

Specific comments

- Methods: In experiment 1, page 3, lines 60, it is said that "the experiment consisted of different combinations... ", referring to the experimental treatments. I would expect all the possible combinations to be shown, in a full crossed design. However, later, in the results (Fig. 2) I see that the combinations of the control diet at high food concentrations is missing for both photoperiods. This will affect the interpretation of the GLM performed, because the factor 'concentration' is not fully crossed. I don't see the reason for this. The authors should provide an explanation, since if *R. subcapitata* could colonize the brood chamber also at high concentrations, that would change the interpretation of results.

Decision letter (RSOS-200249.R0)

31-Mar-2020

Dear Dr Albini,

The editors assigned to your paper ("Turning defence into offence? Intrusion of cladoceran brood chambers by a green alga leads to reproductive failure") have now received comments from reviewers. We would like you to revise your paper in accordance with the referee and Associate Editor suggestions which can be found below (not including confidential reports to the Editor). Please note this decision does not guarantee eventual acceptance.

Please submit a copy of your revised paper before 23-Apr-2020. Please note that the revision deadline will expire at 00.00am on this date. If we do not hear from you within this time then it will be assumed that the paper has been withdrawn. In exceptional circumstances, extensions may be possible if agreed with the Editorial Office in advance. We do not allow multiple rounds of revision so we urge you to make every effort to fully address all of the comments at this stage.

If deemed necessary by the Editors, your manuscript will be sent back to one or more of the original reviewers for assessment. If the original reviewers are not available, we may invite new reviewers.

- Data accessibility

If you wish to submit your supporting data or code to Dryad (<http://datadryad.org/>), or modify your current submission to dryad, please use the following link:
<http://datadryad.org/submit?journalID=RSOS&manu=RSOS-200249>

- Competing interests

- Authors' contributions

- Acknowledgements

- Funding statement

on behalf of Dr Berat Haznedaroglu (Associate Editor) and Pete Smith (Subject Editor)
 openscience@royalsociety.org

Comments to Author:

Reviewers' Comments to Author:

Reviewer: 1

Comments to the Author(s)

This is an interesting work. Adaptations of algae and other phytoplankton groups against grazers such as cladocerans are long known. These include formation of colonies, development of spines, elevated levels of toxin production, thickened cell wall formation, etc. Even without apparent morphological adaptations, simple and non-colonial alga such *Chlorella* can still develop resistance against grazers (Yoshida et al. 2004 Proc. R. Soc. Lond. B 271: 1947–1953). The authors have shown *Chlorella* colonizes the brood chamber/pouch of cladocerans to kill the grazers. This observation as such is not novel. It is known that the brood chamber of cladocerans is of two types: open and close. For many species of cladocerans including *Daphnia*, the brood pouch is open posteroventrally, where it is contact with the medium. From this region, epibiotic microorganisms or those from the medium enter the brood chamber (Smirnov NN 2018. Physiology of the Cladocera 2nd edition, Academic Press). In this case the authors have observed that *Chlorella* enters the brood pouch. The growth of algae in the cladoceran brood pouch is also not entirely unexpected because the medium in the brood pouch maintains osmotic concentration similar to that of the surrounding medium (Aladin & Potts 1995. J Comp Physiol B 164: 671–683). Since the conditions in the brood pouch are not hostile to alga or bacteria, fungi, etc., such organisms can proliferate there.

Reviewer: 2

Comments to the Author(s)

The authors did an interesting job. They analyzed the anti-grazer strategy in the green alga *Chlorella vulgaris* against two cladoceran grazers, *Daphnia magna* and *Simocephalus* sp. They found that *C. vulgaris* could directly inflict harm on the reproductive structure of *D. magna*, and

the colonization process depended on light conditions. To our knowledge, this is indeed the first study demonstrating the harmful intrusion of cladoceran brood chambers by a freshwater microalga, although the underlying mechanism and the specific process has still not been well clarified in the present study. Therefore, I recommend publishing this novel discovery in the journal after a major revision. The specific opinions are as follows.

Major questions

Method section

(1) It is unclear whether this phenomenon could occur in naturally aquatic ecosystems given that the colonization of brood chamber by *Chlorella vulgaris* mainly depended on extremely light conditions. Firstly, the photoperiod throughout their experiments were 18L:6D or even 24 L: 0D, which was not in accordance with the natural condition. This is particularly true when the long-term light intensity ($130 \mu\text{mol photons m}^{-2} \text{ s}^{-1}$) was also considered.

(2) Although with similar cell size, the morphology of the two used algal species was different, is this the reason for their difference in colonization ability? This possibility should be discussed.

(3) The third experiment lasted for 10 days. Is it long enough for individuals of *Simocephalus* to reach their maximum body size? If not, your conclusion that *Simocephalus* individuals cannot be colonization by *Chlorella vulgaris* is not rigorous. Please explain.

Discussion section

(4) You suggest that the success of the *C. vulgaris* cells to remain inside the brood chambers was associated with its ability in high production of EPS under higher light availability? How do you know *R. subcapitata* do not have this ability? In other words, why *R. subcapitata* cannot invade grazers brood chamber?

(5) In addition, the underlying mechanisms for the failure of *C. vulgaris* cells to invade brood chamber of *Simocephalus* should also be discussed.

(6) Any microbial process facilitate the intrusion?

Minor questions.

1. Line 4-7, Page 3, the experimental procedure is not clear. For example, the 20 ml tube contain 17 ml Evian spring water, is oxygen availability of cladoceran grazers limited by this treatments? Was the tube capped with breathable polyethylene?

2. Line 13-14, page 4, What kind of magnifying lens was used? Is the magnification large enough to do this work?

3. Line 36-37, page 4, The culture condition for algae should be added here.

4. Line 1-5, Page 5, Did you check the prerequisites for GLM tests?

5. Line 20-38, page 6, It would be better if you could give some pictures to prove your hypothesis about the specific process how *C. vulgaris* cells entrance the brood chamber of *D. magna*.

6. Line 20-38, page 6, Is it possible for *C. vulgaris* cells entrance the brood chamber of *D. magna* from the postabdomen of *D. magna*, such as during the process of discharging neonates.

7. For figure 2 and 3, it is not clear what data you used for these two figures. Is it at the end of the experiment on a specific day?

Reviewer: 3

Comments to the Author(s)

Review of Manuscript RSOS-200249

General comments

The results presented in this paper show the ability of one common species of microalga (*Chlorella vulgaris*) to colonise the brood chambers and induce reproductive failure in two cladoceran species. One of them, of great significance in the ecosystems and the scientific literature (*Daphnia magna*). Colonization of the brood chamber is positively related with longer light-photoperiod and stronger light intensity. There is some issue with the experimental design, that authors should solve, and its detailed in the specific comments below.

I miss some discussion about an ecological interpretation (beyond that high light conditions increase EPS production by *C. vulgaris*) of the potential significance of this mechanism of stronger exposure to light in the real ecosystems.

The authors claim that this is the first time that an interaction of this nature was found. However, something similar was found between two unicellular planktonic organisms: the haptophyte *Prymnesium parvum* and the dinoflagellate *Oxhyrris marina* (Tillmann et al. 2003, *Aquatic Microbial Ecology* 34: 73-84). This dinoflagellate is a heterotrophic organism that usually phagocytoses phytoplankton cells, like *P. parvum*. *P. parvum* was observed to release cytolytic toxins that lyse *O. marina* cells, to phagocytose the small remains of the cells. This interaction is different than the one reported here because the predator actually becomes the prey, however, I think it needs to be referenced.

I would recommend this paper for publication after the authors solve the few questions presented here.

Specific comments

- Methods: In experiment 1, page 3, lines 60, it is said that “the experiment consisted of different combinations...”, referring to the experimental treatments. I would expect all the possible combinations to be shown, in a full crossed design. However, later, in the results (Fig. 2) I see that the combinations of the control diet at high food concentrations is missing for both photoperiods. This will affect the interpretation of the GLM performed, because the factor ‘concentration’ is not fully crossed. I don’t see the reason for this. The authors should provide an explanation, since if *R. subcapitata* could colonize the brood chamber also at high concentrations, that would change the interpretation of results.

Author's Response to Decision Letter for (RSOS-200249.R0)

See Appendix A.

RSOS-200249.R1 (Revision)

Review form: Reviewer 1

Is the manuscript scientifically sound in its present form?

Yes

Are the interpretations and conclusions justified by the results?

Yes

Is the language acceptable?

Yes

Do you have any ethical concerns with this paper?

No

Have you any concerns about statistical analyses in this paper?

Yes

Recommendation?

Accept with minor revision (please list in comments)

Comments to the Author(s)

Can the figures 2 & B be tested statistically using multiple comparisons?

Review form: Reviewer 3

Is the manuscript scientifically sound in its present form?

Yes

Are the interpretations and conclusions justified by the results?

Yes

Is the language acceptable?

Yes

Do you have any ethical concerns with this paper?

No

Have you any concerns about statistical analyses in this paper?

No

Recommendation?

Accept as is

Comments to the Author(s)

I am satisfied with the revision that the authors made according to my previous comments. My main issue with this work was about the lack of a fully crossed experimental design, but now I understand that this was due to practical reasons, and also see that the "high food" level was tested in the second experiment. Although this is not exactly equivalent to a fully crossed design, it is acceptable, and authors are including this aspect in the discussion.

Decision letter (RSOS-200249.R1)

Dear Dr Albini,

On behalf of the Editors, I am pleased to inform you that your Manuscript RSOS-200249.R1 entitled "Turning defence into offence? Intrusion of cladoceran brood chambers by a green alga leads to reproductive failure" has been accepted for publication in Royal Society Open Science subject to minor revision in accordance with the referee suggestions. Please find the referees' comments at the end of this email.

The reviewers and Subject Editor have recommended publication, but also suggest some minor revisions to your manuscript. Therefore, I invite you to respond to the comments and revise your manuscript.

- Ethics statement

- Data accessibility

<http://datadryad.org/submit?journalID=RSOS&manu=RSOS-200249.R1>

- Competing interests

- Authors' contributions

- Acknowledgements

- Funding statement

Because the schedule for publication is very tight, it is a condition of publication that you submit the revised version of your manuscript before 05-Aug-2020. Please note that the revision deadline will expire at 00.00am on this date. If you do not think you will be able to meet this date please let me know immediately.

Best regards,

on behalf of Dr Berat Haznedaroglu (Associate Editor) and Pete Smith (Subject Editor)
openscience@royalsociety.org

Subject editor comments:

There is a request from Reviewer 1 to conduct a statistical analysis for the Figures. Please consider this useful suggestion.

Reviewer comments to Author:

Reviewer: 1

Comments to the Author(s)

Can the figures 2 & B be tested statistically using multiple comparisons?

Reviewer: 3

Comments to the Author(s)

I am satisfied with the revision that the authors made according to my previous comments. My main issue with this work was about the lack of a fully crossed experimental design, but now I understand that this was due to practical reasons, and also see that the "high food" level was tested in the second experiment. Although this is not exactly equivalent to a fully crossed design, it is acceptable, and authors are including this aspect in the discussion.

Author's Response to Decision Letter for (RSOS-200249.R1)

See Appendix B.

Decision letter (RSOS-200249.R2)

Dear Dr Albini,

It is a pleasure to accept your manuscript entitled "Turning defence into offence? Intrusion of cladoceran brood chambers by a green alga leads to reproductive failure" in its current form for publication in Royal Society Open Science.

on behalf of Dr Berat Haznedaroglu (Associate Editor) and Pete Smith (Subject Editor)
openscience@royalsociety.org

Appendix A

Dear Editor,

We are submitting a revised version of our manuscript “Turning defence into offence? Intrusion of cladoceran brood chambers by a green alga leads to reproductive failure.” for consideration in *Royal Society Open Science*.

We have made every effort to fully address all of the reviewer comments. In particular, we have added new information to clarify the relevance of our experimental conditions to the light conditions experienced by *Chlorella vulgaris* within its natural range, how morphological differences between the algae species may have contributed to our results, further details about the Methodology, and how robust our interpretations are to the experimental design. Detailed point-by-point responses are given below, with our responses formatted in blue.

We believe the revisions we have made have improved the manuscript and are grateful to the reviewers and Editorial team for their feedback. We look forward to hearing back from you.

On behalf of all authors,
Dr Dania Albini

Response to Referees

Reviewer: 1

This is an interesting work. Adaptations of algae and other phytoplankton groups against grazers such as cladocerans are long known. These include formation of colonies, development of spines, elevated levels of toxin production, thickened cell wall formation, etc. Even without apparent morphological adaptations, simple and non-colonial alga such *Chlorella* can still develop resistance against grazers (Yoshida et al. 2004 Proc. R. Soc. Lond. B 271: 1947–1953). The authors have shown *Chlorella* colonizes the brood chamber/pouch of cladocerans to kill the grazers. This observation as such is not novel. It is known that the brood chamber of cladocerans is of two types: open and close. For many species of cladocerans including *Daphnia*, the brood pouch is open posteroventrally, where it is contact with the medium. From this region, epibiotic microorganisms or those from the medium entre the brood chamber (Smirnov NN 2018. Physiology of the Cladocera 2nd edition, Academic Press). In this case the authors have observed that *Chlorella* enters the brood pouch. The growth of algae in the cladoceran brood pouch is also not entirely unexpected because the medium in the brood pouch maintains osmotic concentration similar to that of the surrounding medium (Aladin & Potts 1995. J Comp Physiol B 164: 671-683). Since the conditions in the brood pouch are not hostile to alga or bacteria, fungi, etc., such organisms can proliferate there.

REPLY: We thank the reviewer for acknowledging that the article is interesting. We would like to reiterate that the novelty of our study lies not in the fact that *Daphnia* brood chamber is open to the external environment (as it has been elegantly demonstrated by Seidl et al. (2002) in their study of the brood current), or that pathogens and parasites could exploit the opportunity to inflict harm on *Daphnia*; rather, there is no precedent where a microalgal food species enters the brood chamber, colonises the eggs and causes reproductive failure. While some may expect

such a phenomenon to occur, we now provide the first scientific evidence to support such an expectation. Nevertheless, we thank the reviewer for the very interesting comments and suggestions, which we have now incorporated into the Introduction and Discussion. (P2 L18-19; P6 L2-5)

Reviewer 2:

The authors did an interesting job. They analyzed the anti-grazer strategy in the green alga *Chlorella vulgaris* against two cladoceran grazers, *Daphnia magna* and *Simocephalus* sp. They found that *C. vulgaris* could directly inflict harm on the reproductive structure of *D. magna*, and the colonization process depended on light conditions. To our knowledge, this is indeed the first study demonstrating the harmful intrusion of cladoceran brood chambers by a freshwater microalga, although the underlying mechanism and the specific process has still not been well clarified in the present study. Therefore, I recommend publishing this novel discovery in the journal after a major revision. The specific opinions are as follows.

REPLY: We thank the reviewer for acknowledging that the findings are novel and the article is interesting, and for suggestions to improve the manuscript.

Major question

Method section

(1) It is unclear whether this phenomenon could occur in naturally aquatic ecosystems given that the colonization of brood chamber by *Chlorella vulgaris* mainly depended on extremely light conditions. Firstly, the photoperiod throughout their experiments were 18L:6D or even 24 L: 0D, which was not in accordance with the natural condition. This is particularly true when the long-term light intensity ($130 \mu\text{mol photons m}^{-2} \text{s}^{-1}$) was also considered.

REPLY: *C. vulgaris* has a global distribution, stretching from the arctic (Matula et al 2007) to the tropics (Pham et al 2011), which suggests that the photoperiods and light intensities used in our experiments will be found within their natural range. This point has now been added to the Introduction (P2 L27 - 30). Further, the light intensity we used is lower than the incipient light intensity and the mean surface mixed-layer light intensity of many lakes (e.g. Pérez et al. 2002, Staehr et al. 2016). We first used 24:0 L:D photoperiod, admittedly a rather extreme condition, to test for the occurrence of the new trait in *Chlorella vulgaris*, but we did not stop there. We repeated the experiment using the more common photoperiod of 18:6 L:D (which is a common photoperiod in summer across *Chlorella*'s distribution) with a higher light intensity ($\sim 130 \mu\text{mol photons m}^{-2} \text{s}^{-1}$) and we showed the phenomenon also occurred in this case. These experiments, together, allow us to discover a novel predator-prey interaction in the plankton that warrants further investigation. Going beyond natural ecosystems, given the fact that *Chlorella* is widely used to feed cladocerans in experimental and aquaculture systems, our findings should prompt reconsideration of the practices as well. (P5 L54 – P6 L5)

(2) Although with similar cell size, the morphology of the two used algal species was different, is this the reason for their difference in colonization ability? This possibility should be discussed.

REPLY: We have discussed this possibility in the Discussion section (P5 L10 - 14):

“Both *C. vulgaris* and *R. subcapitata* cells were less than 12 µm in the longest dimension, whereas the gap when *D. magna* and *Simocephalus* sp. opened their abdominal processes was 577 ± 7.05 µm (mean \pm s.d., n = 50; Figure S3) and 500 ± 33 µm (mean \pm s.d., n = 50), respectively. Hence, the gap was clearly large enough for both algal species to enter the brood chamber via the brood current.”

Therefore, size alone cannot explain the different outcomes. If by “morphology” the reviewer was referring to cell shape, neither species have spiny structures or protrusions that may suggest entanglement inside the brood chamber. We were indeed surprised by our findings which suggest other interspecific differences leading to different outcomes between the two species. We have now also added the following (P5L23-27):

“Venancio et al. (2017) showed that *C. vulgaris* and *R. subcapitata* are very different in their abilities to survive under different environmental factors. In fact, the first can survive under a wider range of ionic conditions compared to the latter [25]. Therefore, *C. vulgaris* has previously been shown to have a competitive advantage over *R. subcapitata*.”

We hope that publication of these results will encourage others to build on this study to investigate the question further.

(3) The third experiment lasted for 10 days. Is it long enough for individuals of *Simocephalus* to reach their maximum body size? If not, your conclusion that *Simocephalus* individuals cannot be colonization by *Chlorella vulgaris* is not rigorous. Please explain.

REPLY: Perhaps the reviewer may have confused the species we tested here. Between the two algal species *R. subcapitata* and *C. vulgaris*, only *C. vulgaris* showed brood chamber colonisation. However, both of the cladoceran species *D. magna* and *Simocephalus* sp. were colonised by *C. vulgaris*. Please refer to the third experiment where we showed brood chamber colonization of *Simocephalus* by *Chlorella vulgaris*, starting on day 7, where 33 out of 40 females were colonised (Figure 1C). The fact that we demonstrated brood chamber colonisation with a second cladoceran species allows us to conclude that the phenomenon was not unique to *D. magna*.

Discussion section

(4) You suggest that the success of the *C. vulgaris* cells to remain inside the brood chambers was associated with its ability in high production of EPS under higher light availability? How do you know *R. subcapitata* do not have this ability? In other words, why *R. subcapitata* cannot invade grazers brood chamber?

REPLY: We can only speculate that the production of EPS was the reason based on the literature (P5 L30-40). Further experiments with both algae species would aim to discover the nature of these substances and to characterise it, to better understand this phenomenon. Please also see our response to question (2) above.

(5) In addition, the underlying mechanisms for the failure of *C. vulgaris* cells to invade brood chamber of *Simocephalus* should also be discussed.

REPLY: As mentioned above (and in the original submission), we confirmed that *C. vulgaris* did invade and colonise the brood chamber of *Simocephalus*. Please refer to the third experiment, Figure 1C and our response to question (3) above.

(6) Any microbial process facilitate the intrusion?

REPLY: Further experiments have to be made in order to understand the exact process that guides *Chlorella* into the brood chamber of the predators, and potential involvement of microbial processes. Nevertheless, Seidl et al. (2002) have demonstrated that *Daphnia* creates a “brood current” which pumps water through the posterior into the brood chamber and out through small gaps in the ventral carapace of the cladocerans. We suggest that it can be responsible for bringing *C. vulgaris* cells into the brood chamber. (P2 10-12, P5 L6-10)

Minor questions.

1. Line 4-7, Page 3, the experimental procedure is not clear. For example, the 20 ml tube contain 17 ml Evian spring water, is oxygen availability of cladoceran grazers limited by this treatments? Was the tube capped with breathable polyethylene?

REPLY: We thank the reviewer for the observation. We have added this information in the Methods section: “The tubes were capped with breathable film in order to allow for gas exchange but to limit evaporation of the water”. (P3 L1-2)

2. Line 13-14, page 4, What kind of magnifying lens was used? Is the magnification large enough to do this work?

REPLY: The magnification of the lens was 40x 25mm. This information is been added to the manuscript (P3 L8-9) Prior to start the experiment, we tested the lens to confirm that it was possible to detect the phenomenon.

3. Line 36-37, page 4, The culture condition for algae should be added here.

REPLY: We have now added this information to the methods section: “Both algae were cultured in BG-11 medium (Sigma-Aldrich, 73816 FLUKA) at $20\pm 1^\circ\text{C}$ under an 18L:6D photoperiod and a light intensity of $70 \mu\text{mol photons m}^{-2} \text{s}^{-1}$.” (P3L40-42)

4. Line 1-5, Page 5, Did you check the prerequisites for GLM tests?

REPLY: Yes, the structure of the response data met the GLM test requirements.

5. Line 20-38, page 6, It would be better if you could give some pictures to prove your hypothesis about the specific process how *C. vulgaris* cells entrance the brood chamber of *D. magna*.

REPLY: The aim of the present study was to demonstrate the phenomenon of brood chamber colonisation by *C. vulgaris* and the effects on reproduction. It was beyond our capacity to include the measurements as suggested by the reviewer. Nevertheless, we agree that pictures or videos of the movements of *C. vulgaris* entering the brood chamber would be beneficial to support our hypothesis and this will be considered in future experiments. We have also highlighted this point in the discussion. (P5 L44)

6. Line 20-38, page 6, Is it possible for *C. vulgaris* cells entrance the brood chamber of *D. magna* from the postabdomen of *D. magna*, such as during the process of discharging neonates.

REPLY: We do not think that was the case because we started the experiments with *D. magna* with an empty brood chamber then we closely followed the colonisation process over time. Please refer to the methods section.

7. For figure 2 and 3, it is not clear what data you used for these two figures. Is it at the end of the experiment on a specific day?

REPLY: Both figures show the data for the overall experimental time. We have updated the Figure captions to clarify this point.

Reviewer: 3

The results presented in this paper show the ability of one common species of microalga (*Chlorella vulgaris*) to colonise the brood chambers and induce reproductive failure in two cladoceran species. One of them, of great significance in the ecosystems and the scientific literature (*Daphnia magna*). Colonization of the brood chamber is positively related with longer light-photoperiod and stronger light intensity.

REPLY: We thank the reviewer for the comment and for acknowledging the importance of our study.

There is some issue with the experimental design, that authors should solve, and its detailed in the specific comments below.

I miss some discussion about an ecological interpretation (beyond that high light conditions increase EPS production by *C. vulgaris*) of the potential significance of this mechanism of stronger exposure to light in the real ecosystems.

REPLY: We thank the reviewer for the suggestion, which we have incorporated into the Discussion. (P5 L61 – P6 L7).

The authors claim that this is the first time that an interaction of this nature was found. However, something similar was found between two unicellular planktonic organisms: the haptophyte *Prymnesium parvum* and the dinoflagellate *Oxhyrris marina* (Tillmann et al. 2003, Aquatic Microbial Ecology 34: 73-84). This dinoflagellate is a heterotrophic organism that usually phagocytes phytoplankton cells, like *P. parvum*. *P. parvum* was observed to release cytolytic toxins that lyse *O. marina* cells, to phagocyte the small remains of the cells. This interaction is different that the one reported here because the predator actually becomes the prey, however, I think it needs to be referenced.

REPLY: We thank the reviewer for the suggestion and we have referenced Tillmann et al. 2003 in the manuscript. (P1 L37-P2 L8)

I would recommend this paper for publication after the authors solve the few questions presented here.

Specific comments

- Methods: In experiment 1, page 3, lines 60, it is said that “the experiment consisted of different combinations...”, referring to the experimental treatments. I would expect all the possible combinations to be shown, in a full crossed design. However, later, in the results (Fig. 2) I see that the combinations of the control diet at high food concentrations is missing for both photoperiods. This will affect the interpretation of the GLM performed, because the factor ‘concentration’ is not fully crossed. I don’t see the reason for this. The authors should provide an explanation, since if *R*.

subcapitata could colonize the brood chamber also at high concentrations, that would change the interpretation of results.

REPLY: We have added the following point to the Discussion to clarify (P5 L18 - 22):

“While we did not fully cross all possible concentration combinations in Experiment 1 (due to practical and resource limitations), we did explicitly test the higher *R. subcapitata* concentrations (6×10^6 cells ml^{-1}) in Experiment 2 and still did not record any brood chamber colonisation with this algal species. For this reason, we believe our general interpretations to be robust.”

Literature cited:

1. Gonzalo L. Pérez, Claudia P. Queimaliños, Beatriz E. Modenutti. 2002. Light climate and plankton in the deep chlorophyll maxima in North Patagonian Andean lakes, *Journal of Plankton Research*, **24**, 591–599.
2. Peter A. Staehr, Ludmila S. Brighenti, Mark Honti, Jesper Christensen, Kevin C. Rose. 2016. Global patterns of light saturation and photoinhibition of lake primary production. *Inland Water*, **6**, 593-607.

Appendix B

Dear Editor,

We are submitting the revised version of our manuscript “Turning defence into offence? Intrusion of cladoceran brood chambers by a green alga leads to reproductive failure.”

We have addressed the additional reviewer comment.

On behalf of all authors,
Dr Dania Albini

REPLY TO REVIEWERS

Reviewer:
Comments to the Author(s)

Can the figures 2 & B be tested statistically using multiple comparisons?

REPLY: We thank the reviewer for the question. The corresponding results in Table 1 (and other tables) provide sufficient information to interpret these figures without post-hoc analysis: each factor in the additive GLMs consists of only 2-levels (indicated by 1 d.f. in each row of Table 1). Therefore, any significant difference in that factor represents a difference between the two levels of that factor. Given the lack of interaction terms included in our GLMs, it is not appropriate to perform any further pairwise comparisons, as they would not correspond to these statistical models.

We did, however, perform further analyses including the 2-way interaction between Photoperiod x Algal Species where appropriate, but this did not show qualitative differences from the additive GLMs. Further this more complex statistical model did not describe the data more parsimoniously than the simpler additive model we originally considered. As our data are publicly available on Dryad, interested readers will be able to carry out any further analyses as desired.

Therefore, we have modified the Methods (Section 3.5) to clarify this issue:

“Statistical analysis was performed with R-studio version 1.1.419. An additive, factorial generalized linear model (GLM) with a binomial error family tested for the effect of experimental treatments (algal species, initial concentration, photoperiod and/or light intensity) on the probability of colonisation by the algae. The effect of *C. vulgaris* colonisation on the reproductive output of the grazer was assessed using an additive factorial Negative Binomial GLM [24] examining the effects of the same factors listed above on the number of newborns at the end of the experiment. Given the lack of interaction terms in our analyses (in some cases due to lacking a fully crossed experimental design), we did not perform further pairwise comparisons on these statistical models. Each factor in the additive GLMs consists of only two levels (two photoperiods, two light intensities, two algal species, two initial algal concentrations), therefore any significant

differences associated with a factor can be interpreted as a difference between those two factor levels. Inclusion of interaction terms, where appropriate, did not result in better performing models in any case (based on AICc comparison), or qualitatively change our interpretations.”

Reviewer:

Comments to the Author(s)

I am satisfied with the revision that the authors made according to my previous comments. My main issue with this work was about the lack of a fully crossed experimental design, but now I understand that this was due to practical reasons, and also see that the "high food" level was tested in the second experiment. Although this is not exactly equivalent to a fully crossed design, it is acceptable, and authors are including this aspect in the discussion.

REPLY: We thank the reviewer for the acknowledgment that we provided the requested information.